# Two Types of Wear Mechanisms Governing Transition between Mild and Severe Wear in Ti-6Al-4V Alloy during Dry Sliding at Temperatures of 20–250 °C

**DOI:** 10.3390/ma15041416

**Published:** 2022-02-14

**Authors:** Danhu Du, Wenbin Zhang, Jian An

**Affiliations:** 1Key Laboratory of Automobile Materials, Ministry of Education, School of Materials Science and Engineering, Jilin University, Changchun 130025, China; dudanhu@163.com; 2China Construction Eighth Engineering Bureau Co., Ltd., Shanghai 200122, China

**Keywords:** titanium alloys, sliding wear, wear transition, mechanically mixed layer, softening, dynamic recrystallization

## Abstract

Dry wear characteristics and wear mechanisms governing mild-severe wear transition of Ti-6Al-4V alloy were studied during sliding against medium carbon chromium steel (50Cr) in an experimental temperature range of 20–250 °C. At each experimental temperature, wear rate was plotted against applied load, and its variation was broken into two stages according to the difference of slope. Morphologies and contents of worn surfaces were examined by scanning electron microscope and energy dispersive X-ray spectrometer, from which the two stages were identified to correspond to mild and severe wear, respectively. Two types of wear mechanisms that dominated mild-severe wear transition were found, i.e., breakdown of mechanically mixed layer at temperatures of 20 and 50 °C, and severe plastic deformation at temperatures of 100–250 °C. Microstructures and hardness were examined in the subsurfaces, from which severe plastic deformation-dominated mild-severe wear transition was identified to be caused by the softening arising from friction heating-induced dynamic recrystallization. A linear relation between mild-severe wear transition load and experimental temperature was discovered. The intercept of experimental temperature axis 450 °C was obtained by linearly fitting, and it was considered as a critical dynamic recrystallzation temperature for mild-severe wear transition within the temperature range of 100–250 °C.

## 1. Introduction

As a type of two-phase (α + β) titanium alloy, Ti-6Al-4 V alloy possesses a variety of distinguished features such as low density, high tensile and fatigue strength, and excellent corrosion-resistance performance, which makes it an important structural material that has been widely used in aerospace, chemical, energy, biomedical and military industries [1,2,3]. Despite all these advantages, Ti-6Al-4V alloy presents an unexpectedly poor wear performance, just as other titanium alloys. It even exhibits a lower wear resistance than Al-7072 alloy [4]. The unsatisfactory wear performance of titanium alloys greatly limits their further application in tribological fields, although a few of wear components are made of titanium alloys, such as gear, engine valve, and engine connection rod and turbine disk. The reason for low wear resistance of titanium alloys to this day is still a big concern, and it has evoked great interest among tribologists who hope uncover the truth and take pertinent steps to enhance wear properties of titanium alloys [5,6,7].

Numerous efforts have been made to study wear behavior of various titanium alloys. The poor performance of titanium alloys during wear is thought to be related with several aspects such as low shear strength, poor strain-hardening effect, high coefficient of friction and limited protection effect of fragile tribo-layers [8,9,10]. Among them, trio-oxide layer or mechanically mixed layer (MML) that often occurs on the surface during dry sliding wear is mostly the discussed one, especially its positive role in the transition from mild to severe wear. Light alloys such as aluminum alloys and magnesium alloys often exhibit mild and severe wear behavior depending on various sliding conditions [11,12,13]. Mild wear is often associated with wear mechanisms including abrasion, oxidation and delamination, while severe wear is usually related with large-scale spallation of oxide layer or MML, and severe plastic deformation (SPD). From the viewpoint of engineering application, mild wear is considered as a type of acceptable safe wear, but severe wear is thought as a kind of impermissible dangerous wear. Similarly, mild wear and severe wear are also found in titanium alloys under many dry sliding conditions. As a result, mild-severe wear (henceforth SW for short) transition of titanium alloys has been looked upon as an important issue in engineering applications, and it has attracted lot of attentions of tribologists.

Tribo-oxide layers or MMLs are thought to exercise a strong influence on titanium alloys maintaining mild wear at both room and high temperatures. It is because tribo-oxide layers are prone to form on the surfaces via interaction between O, Ti and Fe elements as well as mechanically mixed debris during dry sliding. Therefore, the composition, hardness and integrity of tribo-oxide layers are regarded as important factors influencing wear resistance and SW transition of titanium alloys. In case of Ti-6.5Al-3.5Mo-1.5Zr-0.3Si alloy sliding at loads of 20–50 N, when more tribo-oxides (e.g., Fe_2_O_3_) were formed in tribo-layers, wear rate was lower and mild wear could keep within a wide range of 1–4 m/s, otherwise severe wear emerged within a narrow range of 1–2 m/s [14]. During Ti-6Al-4V alloy sliding at 1.0 m/s under loads of 50–250 N, wear resistance was found better at higher temperatures of 400 and 500 °C than at lower temperatures of 20 and 200 °C, which was attributed to the higher hardness of tribo-layers formed at 400 and 500 °C [15]. Similarly, during sliding at 1.0 m/s and at high loads of 150–250 N, Ti-6Al-4V and Ti-6.5Al-3.5Mo-1.5Zr-0.3Si alloys were reported to undergo a sudden decrease of wear rate at 400 and 300 °C i.e., a transition from severe to mild wear, which was ascribed to high hardness of tribo-layer containing more Fe_2_O_3_ [16]. In these references [14,15,16,17], the emergence of severe wear in titanium alloys was thought to be due to breaking down or delaminating of tribo-oxide layers. However, there exists another viewpoint that sliding wear behavior is strongly influenced by frictional heating. The point is that frictional heating plays a crucial role because it can affect tribo-oxidation, adiabatic shear banding, MML and softening of material during dry sliding wear of Ti-6Al-4V alloy pin against SS316L disk at 2.8 MPa contact pressure [18]. Clearly, the frictional heating can promote surface plastic deformation and microstructure transformation besides tribo-layer formation even at room temperature. At elevated temperatures, friction force makes it easier too for great changes of microstructure and properties in surface material, which can destroy tribo-layers integrity, but also induce SW transition. These effects have already been reported in aluminum and magnesium alloys [11,12,13]. So far, other SW transition mechanisms for titanium alloys have been rarely reported by available research works except for the tribo-layer breakdown-dominated mechanism. Therefore, a comprehensive study on frictional heating-affected aspects, such as surface morphological development and changes of microstructure and properties in subsurfaces, may be an effective approach of revealing SW transition mechanisms for titanium alloys at elevated temperatures.

The present work is aimed at revealing SW transition mechanisms operating in the temperature range of 20–250 °C. Wear rate variations with applied load were analyzed, and morphologies and chemical compositions of worn surfaces were examined, from which mild and severe wear were separated on a wear mechanism map, and the critical loads for SW transition were determined. By comparison of differences in structure and hardness nearby surface before and after onset of severe wear, two types of wear mechanisms governing SW transition were determined.

## 2. Experimental Details

### 2.1. Material

Ti-6Al-4V alloy bar was received in a hot-rolled state. The cross and longitudinal sectional microstructures were observed using a LEXT-OLS confocal scanning laser microscope (CSLM, Olympus Corporation, Tokyo, Japan), as shown in Figure 1a,b, respectively. The alloy displayed an equiaxed-grain structure of α-Ti phase surround by network-like β-Ti phase on the cross section, while it showed an elongated grain structure of α-Ti phase together with network-like β-Ti phase on the longitudinal section. The average size of α-Ti grains were measured on the cross section, while the average length of α-Ti grains were measured along elongation direction on the longitudinal section. Both of them were measured by the linear intercept method using the equation *d = 1.74 L*, where *L* is the linear intercept size or length. The average size and length of α-Ti grains were around 10.7 μm and 14.5 μm, respectively.

### 2.2. Wear Tests

Pin-on-disk wear tests were carried out using Ti-6Al-4V alloy pins sliding against medium carbon chromium steel (50Cr) disks. The used tribometer was a MG2000 type high-temperature pin-on-disk machine (Chengxin Test Equipment Manufacturing Company Limited, Zhangjiakou, China). Experiments were run at a constant speed of 0.5 m/s within a load range of 10–240 N and within a temperature range of 20–250 °C. Pin specimens were wire-electrical-discharge machined from as-received bar into cylinders with dimensions of ϕ6 mm × 13 mm. The counterface disks have the dimensions of ϕ70 mm × 10 mm, and they were heat treated to a hardness of about 54HRC. Before wear testing, both the pins and disks were ground with 1000# sandpaper and polished on their flat surfaces up to about 0.5 μm surface roughness. The motion mode between pin and disk was that the disk rotated at the chosen speed while pin was contacted with disk under applied load. The sliding track on the disk was a circle with 60 mm diameter. For all wear tests, the sliding distance was 564.7 m i.e., 3000 cycles. When elevated-temperature wear tests were conducted, the friction pairs were kept at an experimental temperature for 5 min in a resistance furnace that was equipped on the wear tester. The required test temperature could be guaranteed with an error of ±5 °C by a thermocouple probe placed in the furnace chamber. After wear testing, the length reduction of pin (μm) was measured using a high-accuracy digital micrometer with ±1 μm deviation, then the volumetric loss of pin could be obtained based on the length reduction and normal area of pin. At each test temperature, wear performance of studied alloy was characterized by wear rate variation with applied load. Wear rate was reported in terms of volumetric loss of material per unit sliding distance (m^3^m^−1^). Each wear rate was acquired from the average through at least three repeated wear tests.

Worn surface morphologies and contents of primary elements (e.g., O, Al, Ti and Fe) were analyzed to identify wear mechanisms using a TESCAN VEGA3 scanning electron microscope (SEM, TESCAN, Brno, Czech Republic) equipped with an energy dispersive X-ray spectrometer (EDS, Oxford Instrument Company, Oxford, UK). The constituent phases on the worn surfaces were examined by a Rigaku D/MAX 2500PC X-ray diffractometer (XRD, Tokyo, Japan) using Cu K_α_ radiation over the 2*θ* range of 20–100° at time per step of 1.25 s and step size of 0.05° under condition of 40 kV and 30 mA.

### 2.3. Microstructure Observation and Hardness Measurements in Subsurfaces

The changes of microstructure and hardness in the near surface region during wear testing were characterized by CLSM and HVS-1000 Vickers microhardness tester (Huayin Test Instrument Company Limited, Laizhou, China). Subsurface specimens were prepared by cutting worn pin into two along a plane that was vertical to the worn surface and parallel to the sliding direction, mounting one in resin, grinding and polishing per metallographic method. Microhardness in subsurfaces was measured with 50 g force for 10 s along the depth direction beneath worn surfaces. At each depth away from surface, microhardness was obtained from the average of measurements at six locations.

## 3. Results and Discussion

### 3.1. Wear at 20 and 50 °C

The wear rates at 20 and 50 °C were plotted against applied load, as illustrated in Figure 2. The two wear rate vs. applied load curves exhibited a similar increasing trend with load except for two parts with lower wear rate at 50 °C within 80–140 N and within 160–220 N, respectively. Both wear rates increased gradually within 10–60 N and rose up a little at 80 N, and then maintained a low slope within 80–120 N and within 80–140 N, respectively. According to the stage characteristic, the parts of wear-rate curves within 10–120 N at 20 °C and within 10–140 N at 50 °C were defined as the first stages, in which wear rates increased gradually and were generally in a low level. Thereafter, wear rates began to go up to a high level around 45 × 10^−12^ m^3^m^−1^ and kept that status within 160–200 N, finally ascended steeply to 60 × 10^−12^ m^3^m^−1^ at 240 N. The other parts of wear-rate curves within 140–240 N at 20 °C and within 160–240 N at 50 °C were defined as the second stages, in which wear rates increased rapidly with load and kept at a high level. In the first stages, the wear rates were found to be below 35 × 10^−12^ m^3^m^−1^. Such a wear rate limit value in the first stages is quite similar to the wear rate at SW transition state for pure titanium pin sliding against hardened gauge steel with hardness 701HV5 at 0.5 m/s, i.e., 30 × 10^−12^ m^3^m^−1^ [19], and it is also approximately equal to the wear rates at SW transition state at 0.5 m/s for several Mg alloys including AZ91, Mg97Zn1Y2 and Mg-Gd-Y-Zr alloys, e.g., 39.9 × 10^−12^ m^3^m^−1^ for AZ91 alloy, 39.8 × 10^−12^ m^3^m^−1^ for Mg97Zn1Y2 alloy, 31.7 × 10^−12^ m^3^m^−1^ for Mg-10.1Gd-1.4Y-0.4Zr alloy [12,20,21]. Therefore, it can be preliminarily assumed that the wear is mild in the first stage, but it is severe in the second stage, and the corresponding loads for SW transition at 20 and 50 °C are 120 N and 140 N, respectively.

Worn surface morphologies and chemical compositions at 20 and 50 °C were examined for identifying wear mechanisms. Figure 3 shows SEM micrographs of surfaces, while Table 1 lists the primary element contents. The wear mechanisms at 20 and 50 °C were oxidation + abration and spallation of MML in the first stages, and breakdown of MML + mild plastic deformation (MPD) and SPD in the second stages. The load ranges corresponding to various mechanisms in the first and second stages are summarized in Table 2. In the first stages, at 20 N, the two worn surfaces at 20 and 50 °C were characterized by fine powders and shallow grooves formed along the sliding direction (Figure 3a,b), meanwhile oxygen element contents were 8.89% and 8.61%, respectively. These exhibit typical characteristics of oxidation wear and abrasion wear. At higher loads of 80 N (20 °C) and 100 N (50 °C), there were many plates of MML detached off from the worn surfaces, which left behind several shallow spallation scars (Figure 3c) and localized areas covered with uncompacted MML (Figure 3d). It was also noted that the content of Fe element was rather high besides high content of O element, especially for the sliding at 50 °C. Fe content reached 10.34% at 50 °C, much higher than Fe content at 20 °C, i.e., 4.22%. It indicates that MML is fully formed on the worn surface at 50 °C, and it helps wear rate maintain a plateau within 80–140 N, and consequently the wear rate is a little lower at 50 °C than at 20 °C within 80–140 N. XRD analysis was carried out on the above-mentioned two worn surfaces for identifying the types of oxides, as shown in Figure 4. Titanium oxide TiO and iron oxide (magnetite) were confirmed to form on the two worn surfaces. The overall surface damage is mild in the first stages, and these wear mechanisms also usually operate in mild wear for Al-based alloys and Mg-based alloys under room and elevated-temperature sliding conditions [12,13,20].

In the second stages, the surfaces at 20 and 50 °C under 160 N were suffered from MPD, which led to a flat surface appearance. Furthermore, most parts of MML were removed from the surface in big-sized flakes due to its increasing thickness and high fragility. The cracks of MML and big-sized scar were observed on the surfaces (Figure 3e,f), which suggests that a serious surface damage is made due to the breakdown of MML. Thus, wear was severe in the second stages. For the sliding at 50 °C, Fe content decreased to 6.64% at 160 N and even to 2.22% at 180 N, which could be a reason for low stability of MML at higher loads. However, even a large-scale breakdown of MML happened at 160 N, surface material was not suffered from SPD, since no typical extruded edge was formed, as can be seen from the insert in Figure 3f. The surface deformation extent was thus still mild, and the wear mechanisms were MPD + breakdown of MML. At higher load of 240 N, the worn surface at 20 and 50 °C were subjected to SPD, which brought about extrusion of surface material out of the surface, as can be seen at the edge of pins (Figure 3g,h). Furthermore, the MML state on surfaces deteriorated as temperatures was increased from 20 to 50 °C, since the content of Fe element decreased from 3.82% to 1.11%. From the surface damage development in the second stage, it can be concluded that SW transition at 20 and 50 °C is governed by the breakdown of MML, and severe wear become much more serious as subsequent SPD takes place.

### 3.2. Wear at Temperatures of 100–250 °C

The wear rates at 100, 150, 200 and 250 °C were plotted against applied load, as shown in Figure 5. It can be observed that the wear rates at 150 and 200 °C are higher than that at 100 °C under most applied loads, even though the difference is not very significant. However, at the highest experimental temperature of 250 °C, the effect of temperature on wear rate become most distinguishing, since the corresponding wear rate curve is apparently in the highest position within 60–140 N. The four wear rate-applied load curves can also be broken into two stages according to the slope difference and wear rate value level. In the first stages, wear rates essentially increased with load in a low slope or maintained a plateau state except for a little rising at certain loads e.g., 80 N at 100 °C, 60 N at 150 °C and 40 N at 200 and 250 °C, and they were all at low levels below 37 × 10^−12^ m^3^m^−1^, whereas in the second stages, wear rates began to increase fast with increasing load, soon reached a high level of 42–46 × 10^−12^ m^3^m^−1^ and finally went up to their maximums above 51 × 10^−12^ m^3^m^−1^. Even though a few of plateaus also occurred within 140–180 N at 150 °C and within 130–140 N 200 °C in the second stages, their wear rates reached as high as 48 × 10^−12^ m^3^m^−1^ and 42 × 10^−12^ m^3^m^−1^, suggesting that they were all in severe wear. The first stages at 100, 150, 200 and 250 °C corresponded to the load ranges of 10–140 N, 10–120 N, 10–100 N and 10–80 N, respectively, while the second stages corresponded to the load ranges of 160–200 N, 140–180 N, 110–180 N and 100–140 N, respectively. Therefore, it can also be preliminarily concluded that the wear is mild in the first stage, but it is severe in the second stage, and the corresponding loads for SW transition at 100, 150, 200 and 250 °C are 140, 120, 100 and 80 N, respectively.

The wear mechanisms at 100, 150, 200 and 250 °C were spallation of MML, and spallation of MML + MPD in the first stages, and SPD in the second stages. Figure 6 shows SEM images of surfaces at 150 and 200 °C, while Table 3 lists the contents of primary elements. As experimental temperature was 150 °C, in the first stage, at 20 N, MML was found to be detached out off surface in thin flake (Figure 6a), while O and Fe elements reached high contents of 12.39% and 6.60%, respectively. The wear mechanism was spallation of MML. At 60 N, the morphological feature remained the same (Figure 6b), and the contents of O and Fe elements still kept as high as 11.01% and 6.71%, respectively. As load was increased to 100 N, spallation scars almost disappeared (Figure 6c), and the surface appeared flat due to MPD, which was confirmed by examination of the deformation extent at the specimen edge, as demonstrated in the insert in Figure 6c. However, MML was found to be spalled at localized area form the magnification image (Figure 6d). Meanwhile, O element still kept a high content of 11.94%. The wear mechanisms were thus MPD and localized spallation of MML. The MPD contributed to a little rise of wear rate at 100 N. As observed from surface damage, wear in the first stage was still mild. In the second stage, at 140 N, the surface suffered from SPD (Figure 6e), which gave rise to a flatten surface and an extruded edge (Figure 6f), indicating occurrence of severe wear. Furthermore, it is noticeable that the contents of O and Fe elements considerably decrease to 5.48% and 0.63%, respectively. The surface composition change suggests that SPD of surface material is detrimental to the stability of MML, and MML has little influence on SW transition at this situation. The surface morphological change between the first and second stages at 200 °C was similar to that at 150 °C. For example, the wear mechanisms were MPD and spallation of MML at 60 N in the first stage (Figure 6g), and SPD at 130 N in the second stage (Figure 6h). Therefore, SPD of surface material dominates SW transition at temperatures of 100–250 °C.

After identification of various wear mechanisms at temperatures of 20–250 °C in the first and second stages, a wear mechanism map was built, as shown in Figure 7. The horizontal axis represents experimental temperature, and vertical axis stands for applied load. The map consists of two main parts; the top part represents mild wear, the bottom part stands for severe wear. The two parts are separated by solid line ACD′A′ that is decided by SEM observation of MML breakdown within 20–50 °C and SPD within 100–250 °C. This boundary between mild wear and severe wear regions tells that severe wear onsets when MML breakdown wear mechanism operates at temperatures of 20–50 °C or SPD wear mechanism controls the wear process at temperatures of 100–250 °C. The mild-wear part includes three sub-regions where one or two major wear mechanisms operate. They are oxidation + abrasion, MML spallation, and MML spallation + MPD sub-regions, respectively. The boundary between oxidation + abrasion and MML spallation sub-regions is marked by dashed line BB′, which is confirmed by SEM observation of spallation scars but no evident grooves. The boundary between MML spallation and MML spallation + MPD sub-regions is indicated by dashed line CC′, which is determined by SEM observation of formation of flat surface. The severe-wear part comprises two sub-regions divided by dashed line DD′, namely MML breakdown + MPD and SPD sub-regions. Their boundary is determined by observation of SPD features such as flatten surface and extruded edge.

### 3.3. Surface Hardness

Dry sliding wear is actually a surface behavior between friction pairs, in which a series of complicated processes are involved, such as chemical interaction between environmental atmosphere and surface material, material transfer, property change and microstructure transformation of surface material. All these processes usually turn into two factors that influence the wear mostly, i.e., tribo-oxide layer or MML, and hardness. Although the hardness values measured on the worn surface are not as reliable as those measured on polished surfaces, they can still be used to roughly evaluate the change of surface property that is influenced by MML and/or other factors during wear testing. Therefore, four sliding situations were selected to examine the surface hardness; their corresponding experimental temperatures were 50, 100, 150 and 200 °C, respectively. Figure 8 shows the surface hardness variations with applied load. All surfaces present hardness values above 380 HV after wear testing, much higher than 310 HV of as-received alloy. In addition, it is also found that surface hardness increases with applied load in mild wear except for the sliding at 50 °C and 120 N, and essentially reaches the maximum before SW transition, but dropped immediately after SW transition, and finally reduces to a low level. It is noticeable that the critical loads for a sudden drop of surface hardness are equal to the SW transition loads. This phenomenon suggests a close correlation between surface hardness and SW transition.

For the sliding at 50 °C, worn surface hardness increased with applied load and kept a high level with 10–140 N in mild wear, except for a little lower value at 120 N, then drop suddenly to 412HV at 160 N, but rose again at 180 N, and finally decreased until 423HV at 240 N. The high level of surface hardness within 10–140 N is apparently associated with formation and persistence of MML according to above-mentioned SEM morphological analysis. When MML was broken down within 160–180 N in severe wear, it could exert a negative influence on the surface hardness due to its loose structure. Therefore, the surface hardness suddenly decreased to a very low value at 160 N. The hardness rising at 180 N could be associated with the surface hardening induced by large plastic deformation after removal of most part of loose MML. For the sliding at 100, 150 and 200 °C, the surface hardness varied in a similar way with applied load. Their surface hardness increased with increasing applied load within certain load ranges (e.g., 10–140 N at 100 °C, 10–120 N at 150 °C and 10–100 N at 200 °C) till a hardness peak above 500 HV in mild wear, then dropped suddenly to low levels of 397–475 HV in severe wear. Obviously, MML contributed partially to the high surface hardness in the mild wear, especially for the sliding at 150 °C, because O element content kept a very high level of 11.81–12.39%. However, for the sliding at 100 °C, the surface hardness reached its highest level at 120 N and 140 N when the contents of O element were only 7.20% and 6.25%, lower than 8.60% at 60 N and 9.50% at 100 N. Therefore, MML was not a dominant reason for high surface hardness at 120 N and 140 N. The surface deformation-induced hardening could be a major factor for surface hardness keeping a continuous rising trend in mild wear. As for the surface hardness drop occurred in severe wear, the reason could be related with certain changes that took place in microstructure as well as property nearby the surface.

### 3.4. Microstructures and Hardness in Subsurfaces

In order to clarify the exact reasons for SW transition, the microstructure evolution and property change were examined in subsurfaces of pin specimens worn at 20, 50, 100, 150 and 200 °C before and after SW transition. It was found that after SW transition, the primary microstructure features were large-scale breakdown of MML at 20 and 50 °C, and formation of a refined-microstructure zone nearby the surfaces at 100, 150 and 200 °C. The significant hardness change was the softening phenomenon occurred nearby the surfaces at 100, 150 and 200 °C after SW transition. The subsurface microstructures and hardness of two typical examples sliding at 50 and 150 °C are presented in Figure 9 and Figure 10, respectively.

Figure 9 shows the subsurface microstructures at 50 and 150 °C before and after SW transition. For the sliding at 50 °C, at 100 N in mild wear, surface was almost entirely covered by MML, as shown in Figure 9a, even though spallation of MML operated in this condition. Underneath the MML there was a plastic deformation zone in which α-Ti grains and β-Ti phase networks were elongated towards surface. The plastic deformation reached a depth of about 48.5 μm. At 160 N in severe wear, only a little part of MML survived on the surface due to a large-scale breakdown of MML in this condition, as shown in Figure 9b. Meanwhile, the plastic deformation reached a little bigger depth of about 61.7 μm. Apparently, it is the extensive breakdown of MML that gives rise to severe wear. For the sliding at 150 °C, at 60 N in mild wear, MML still occupied quite a part of surface, and a plastic deformation zone with about 57.3 μm thickness followed beneath MML, as shown in Figure 9c. As can be seen in the magnification photograph of Figure 9d, the subsurface microstructure change was just the elongation of α-Ti grains and β-Ti phase network along the sliding direction. Nevertheless, at 150 N in severe wear, MML took over only a little part of surface, most parts of surface were occupied by exposed Ti-6Al-4V alloy, and a friction-affected zone with about 105.8 μm thickness was formed beneath surface under applied normal load and friction force, as shown in Figure 9e. It is noticeable that the microstructure appears refined near the surface, as shown in the magnification microphotograph of Figure 9f. Specially, a quite amount of fine spherical grains were found to be embed in the deformed α-Ti grains, and most flow lines that consisted of β-Ti phase disappeared or were interrupted, suggesting a type of microstructure transformation took place nearby surface. The refined microstructure zone was about 15.4 μm thick, next to it was a 75.6 μm-thick plastic deformation zone. Since a large plastic deformation strain was typically achieved nearby surface, dynamic recrystallization (DRX) could be induced by substantial frictional heating. The phenomenon of DRX microstructure transformation nearby surface during severe wear has been reported in aluminum and magnesium alloys [11,13,20]. Ti-6Al-4V disk was also found to be DRXed near the surface in severe wear during sliding against 304 stainless steel pin at 0.5 m/s and 30 N [22]. Therefore, it is justified to infer that a certain extent of DRX microstructure transformation promotes the formation of refined microstructure in the present study. Since softening usually accompanies DRX realization, the deduction about DRX realization nearby surface can be further testified by hardness measurement.

Figure 10 shows the microhardness variations within the depth range of 5–200 μm beneath worn surface. Since the effects of surface roughness and MML are eliminated within such a depth range, microhardness values were measured more accurately than those measured on worn surfaces with big surface roughness and MML. For the sliding at 50 °C, at 100 and 160 N, their hardness exhibited a descending trend with the increase of depth, and finally reduced to the original hardness of the studied alloy, i.e., around 310HV, as shown in Figure 10a. In addition, in the near surface region with depth less than 20 μm, high load of 160 N brought about a larger enhancement of hardness than low load of 100 N. Such a hardness variation trend is a typical feature of strain hardening. The fact that strain hardening effect weakens with increasing depth is apparently due to reduction of strain along depth direction. Considering the thin thickness of MML at 160 N, the higher hardness in the near surface region is definitely aroused by larger plastic deformation rather than MML. Therefore, Ti-6Al-4V substrate is strain hardened underneath MML, and it is not responsible for the decrease of hardness on worn surface at 160 N. The real reason is breakdown of MML, which actually results in severe wear. However, when the load was increased above 180 N, SPD wear mechanism began to dominate the wear process. The hardness then exhibited a different variation trend with depth. For example, at 240 N, the hardness was much lower at depth below 20 μm, as compared with that hardness at 100 and 160 N. A softening zone was obviously formed nearby surface, and it was the cause of surface hardness decrease at loads above 180 N.

For the sliding at 150 °C, as can be seen in Figure 10b, at 60 N in mild wear, a significant strain-hardening feature was also found in the subsurface, that is, hardness kept decreasing until 310HV. Nevertheless, at 150 N in severe wear, a softening phenomenon was found in near surface region, i.e., a low level of hardness occurred at depth below 15 μm. The hardness in the softening zone was even lower than that of as-received Ti-6Al-4V alloy. Moreover, the thickness of softening zone was found almost equal to that of refined microstructure zone in Figure 8f. Therefore, the refined microstructure zone is identified as DRX microstructure zone. When the plastic-deformed microstructure nearby surface changes into DRX microstructure, material is softened and severely deformed, consequently mild wear develops into severe wear. Based on above analysis, it can be summed up that at temperatures of 20 and 50 °C, the wear mechanism dominating M-S wear transition is large-scale breakdown of MML, while at temperatures of 100–250 °C, the wear mechanism is the SPD that arises from the softening caused by DRX microstructure transformation.

### 3.5. Relation between SW Transition Load and Experimental Temperature

Figure 11 shows SW transition load variation with experimental temperature. It is found that when SW transition is dominated by SPD wear mechanism in the temperature range of 100–250 °C, SW transition load shows a negative linear relation with experimental temperature. The linearly fitted relation between the transition load and experimental temperature can be expressed as Equation (1).
(1)$$F_{T}=-k(T_{C}-T)$$
where *F_T_* is transition load, *k* is −0.4, i.e., the slope of fitted line, *T_c_* is 450 °C, i.e., the intercept of experimental temperature axis, *T* is the experimental temperature. For those SPD-dominated SW transition in magnesium alloys including Mg97Zn1Y2, AS31 and Mg-10.1Gd-1.4Y-0.4Zr alloys under elevated-temperature sliding conditions, transition load had a similar negative linear relation with temperature, and *T_c_* was deduced to be a critical temperature for DRX transformation in material nearby surface [20,23,24]. A contact surface DRX temperature criterion has been proposed to solve the problem of SW transition load in Mg alloys, as expressed by Equation (2) [13], namely SW transition starts as long as friction-affected surface temperature *T_s_* rises up to critical DRX temperature of the experimental material *T_DRX_*.
(2)$$T_{S}\geq T_{DRX}$$

The linear relation suggests that SW transition in the studied alloy also follows the criterion. *T_c_* in Equation (1) is actually the critical surface DRX temperature of Ti-6Al-4V alloy. Qiu et al. [25] found that softening phenomenon took place on the worn surfaces of Ti-6Al-4V alloy as surface temperature was higher than 487 °C (e.g., 0.42*T_m_*, *T_m_* is the melting temperature of Ti-6Al-4V alloy). The critical contact surface temperature of 450 °C is thought to be reasonable for DRX softening in the present study, since it is close to 487 °C.

On the basis of the relation between bulk surface temperature and sliding condition proposed by Lim and Ashby [26], Liang et al. [13] also put forward Equation (3) to evaluate SW transition load using critical DRX temperature.
(3)$$F_{T}=\frac{\left(T_{DRX}-T_{}\right)}{K_{DRX}\mu v}$$
where *μ* is coefficient of friction, *v* is sliding speed, *K_DRX_* is a constant that is related with the dimensions and thermal properties of friction pairs, and it is described by Equation (4).
(4)$$K_{DRX}=\frac{\alpha l_{b}}{A_{n}K_{mp}}$$
where *α* is thermal distribution fraction of the pin, *l_b_* is the mean heat diffusion distance of pin, *A_n_* is the nominal contact area, *K_mp_* is the thermal conductivity of the pin.

In the present study, at SW transition states under various temperature conditions, there is not big difference between the wear rates, the reduction of pin length is very little as compared with 13 mm of the original length, ranging from 0.59 mm to 0.72 mm. Therefore, *l_b_* and *A_n_* can be approximately regarded as constants. Since SW transition happens at a constant surface temperature of 450 °C, the thermal distribution fraction of the pin can also be considered as a constant. The thermal conductivity also varies little in the range of 20–250 °C, ranging from 7.69–9.39 Wm^−1^K^−1^ [27]. *K_DRX_* is thus taken as an approximate constant. In addition, the coefficient of friction varies little at SW transition states, ranging from 0.63 to 0.65, as given in Table 4. Therefore, SW transition load shows a linear relation with experimental temperature. *K_DRX_* is calculated to be 7.69 using experimental data at 100 °C, i.e., *F_T_* = 140 N, *T_DRX_* = 450 °C, *μ* = 0.65 and *v* = 0.5 m/s. Consequently, the transition loads at other experimental temperature can be calculated using Equation (3). As can be seen from Figure 10, the calculated transition loads correspond well with those measured at 100–250 °C. The transition loads for onset of SPD wear at 20 and 50 °C can also be calculated using their respective coefficients of friction, and the calculated values are also found close to the measured ones, as shown in Figure 10. These results prove that SPD-dominated SW transition at 100–250 °C and SPD wear transition at 20 and 50 °C are well evaluated using critical DRX temperature.

## 4. Conclusions

At each experimental temperature, wear rate essentially kept an increasing trend with applied load, and wear rate-applied load curve can be divided into two stages: a gradually increasing and maintaining plateau stage and a fast increasing and/or maintaining high-level plateau stage. At temperatures of 20–100 °C, experimental temperature showed a little effect on wear rate variation, but its influence became significant at higher temperatures of 150–250 °C.The wear mechanisms such as oxidation + abrasion, spallation of mechanically mixed layer and mild plastic deformation operated in mild wear, while breakdown of mechanically mixed layer and severe plastic deformation worked in severe wear. The breakdown of mechanically mixed layer governed mild-severe wear transition at 20 and 50 °C, while severe plastic deformation dominated mild-severe wear transition at temperatures of 100–250 °C.Severe plastic deformation was activated by the softening arising from dynamic recrystallization microstructure transformation.Mild-severe wear transition load decreased with experimental temperature in the range of 100–250 °C, and a linear relationship was found between them. A critical surface dynamic recrystallization temperature 450 °C was obtained by linearly fitting the relation between transition load and experimental temperature at 100–250 °C.

## Figures and Tables

**Figure 1 materials-15-01416-f001:**
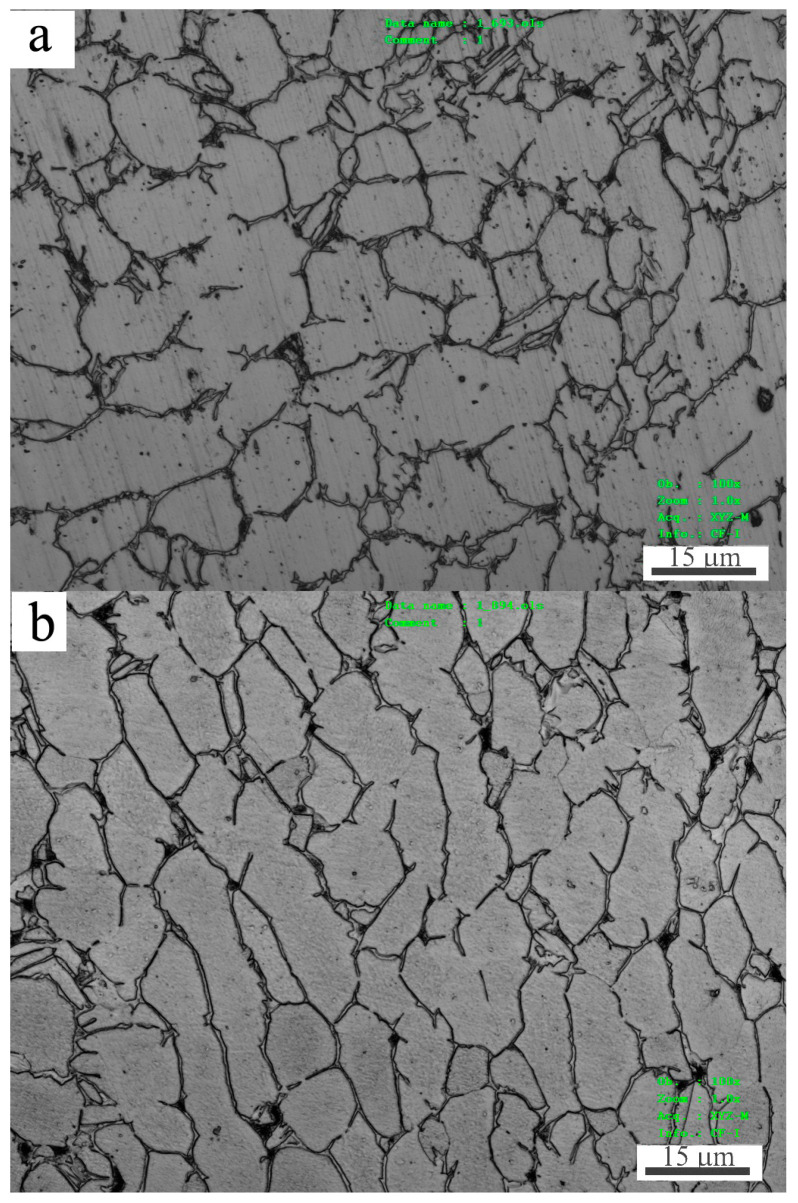
Cross (**a**) and longitudinal (**b**) sectional microstructures of as-received Ti-6Al-4V alloy bar.

**Figure 2 materials-15-01416-f002:**
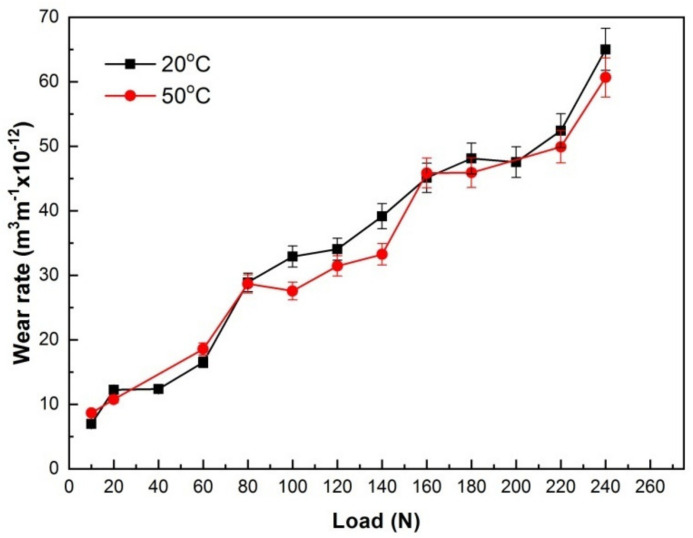
Variations of wear rates with applied load at 20 and 50 °C.

**Figure 3 materials-15-01416-f003:**
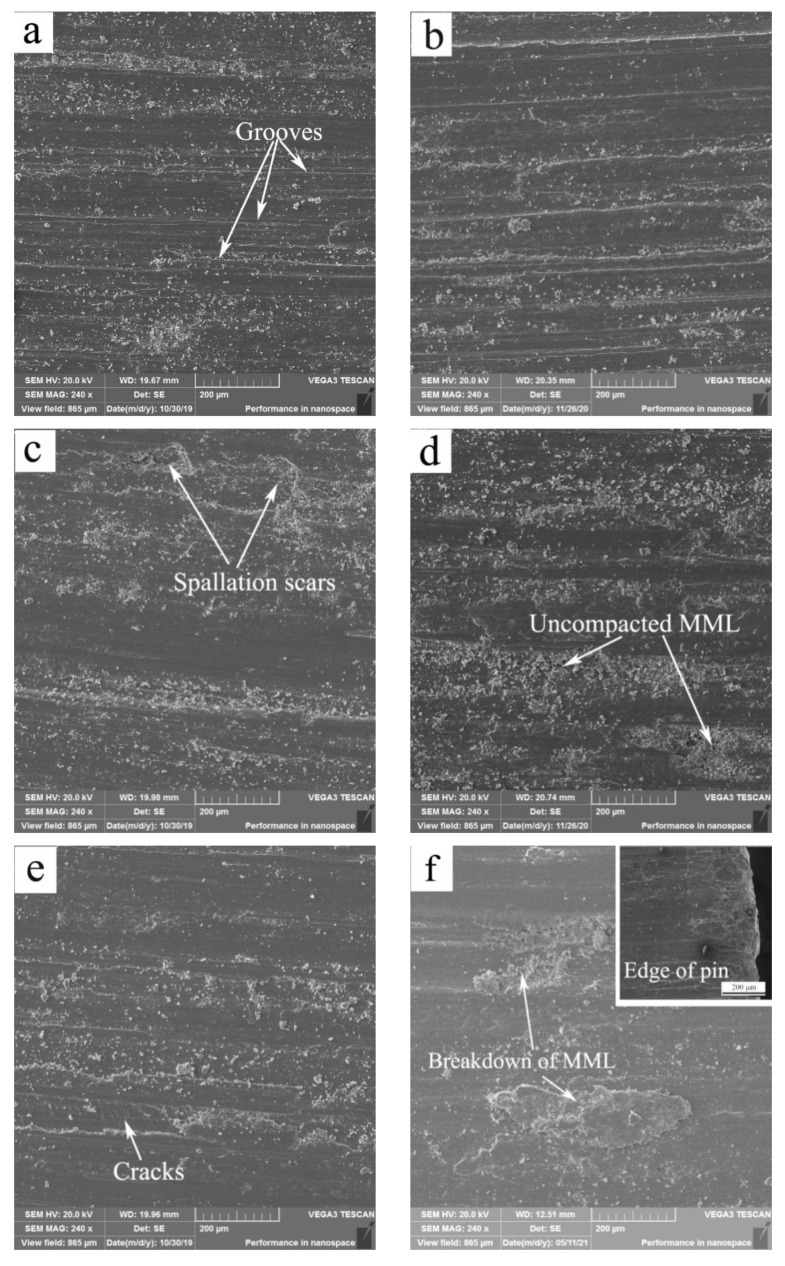
SEM micrographs of worn surfaces at 20 °C (**a**,**c**,**e**,**g**) and 50 °C (**b**,**d**,**f**,**h**) under different loads: (**a**) 20 N, (**b**) 20 N, (**c**) 80 N, (**d**) 100 N, (**e**) 160 N, (**f**) 160 N, (**g**) 240 N, (**h**) 240 N.

**Figure 4 materials-15-01416-f004:**
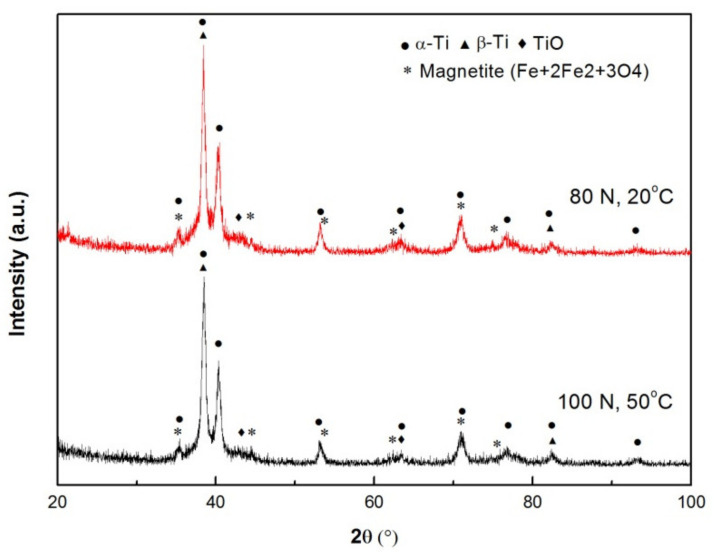
XRD patterns of worn surface at 20 °C and 50 °C.

**Figure 5 materials-15-01416-f005:**
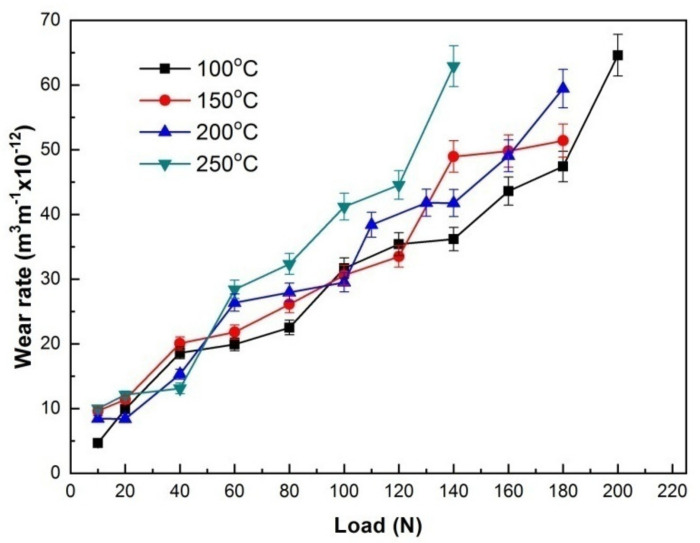
Variations of wear rates with applied load at 100, 150, 200 and 250 °C.

**Figure 6 materials-15-01416-f006:**
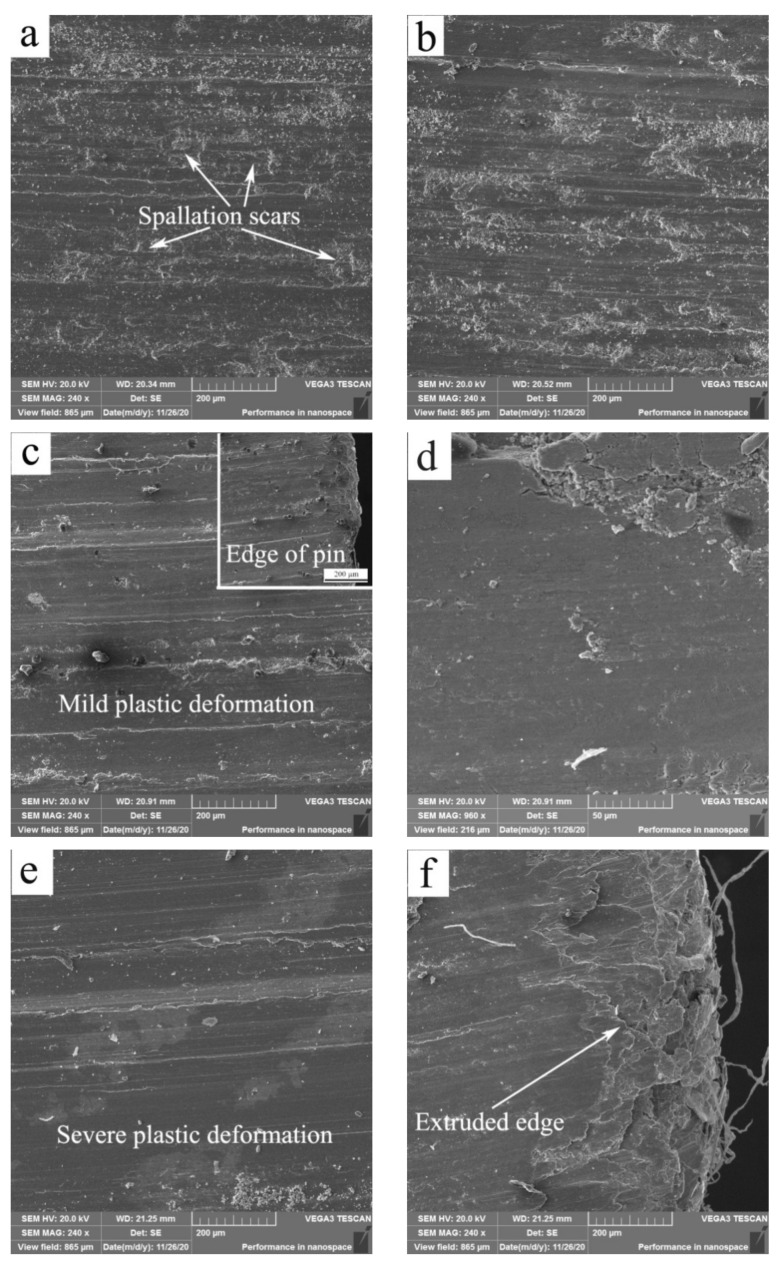
SEM micrographs of worn surfaces at 150 °C (**a**–**f**) and 200 °C (**g**,**h**) under different loads: (**a**) 20 N, (**b**) 60 N, (**c**) 100 N, (**d**) 100 N, (**e**) 140 N, (**f**) 140 N, (**g**) 60 N, (**h**) 130 N.

**Figure 7 materials-15-01416-f007:**
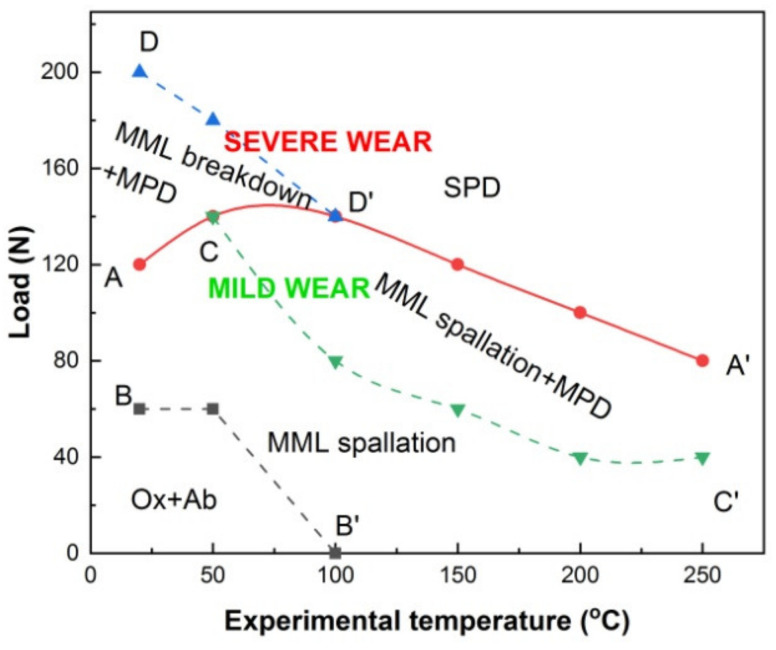
Wear mechanism transition map establish on Experimental temperature-Applied load coordinate system. Ab-Abrasion; Ox-Oxidation; SPD-Severe plastic deformation; MPD-Mild plastic deformation.

**Figure 8 materials-15-01416-f008:**
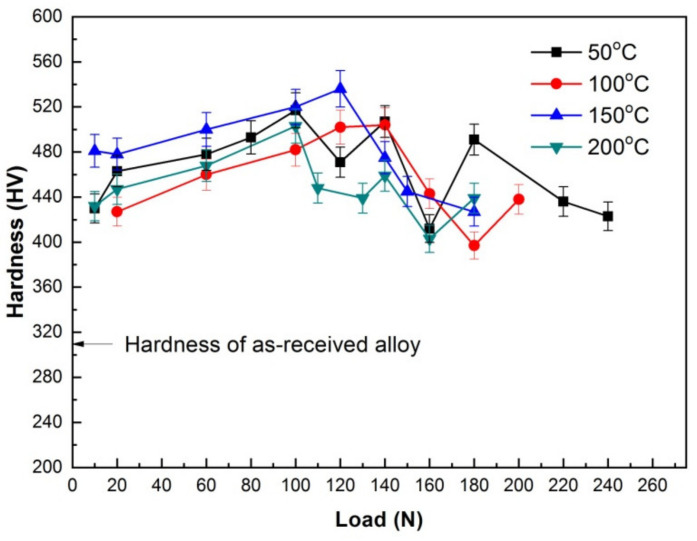
Worn surface hardness vs. applied load at various experimental temperatures.

**Figure 9 materials-15-01416-f009:**
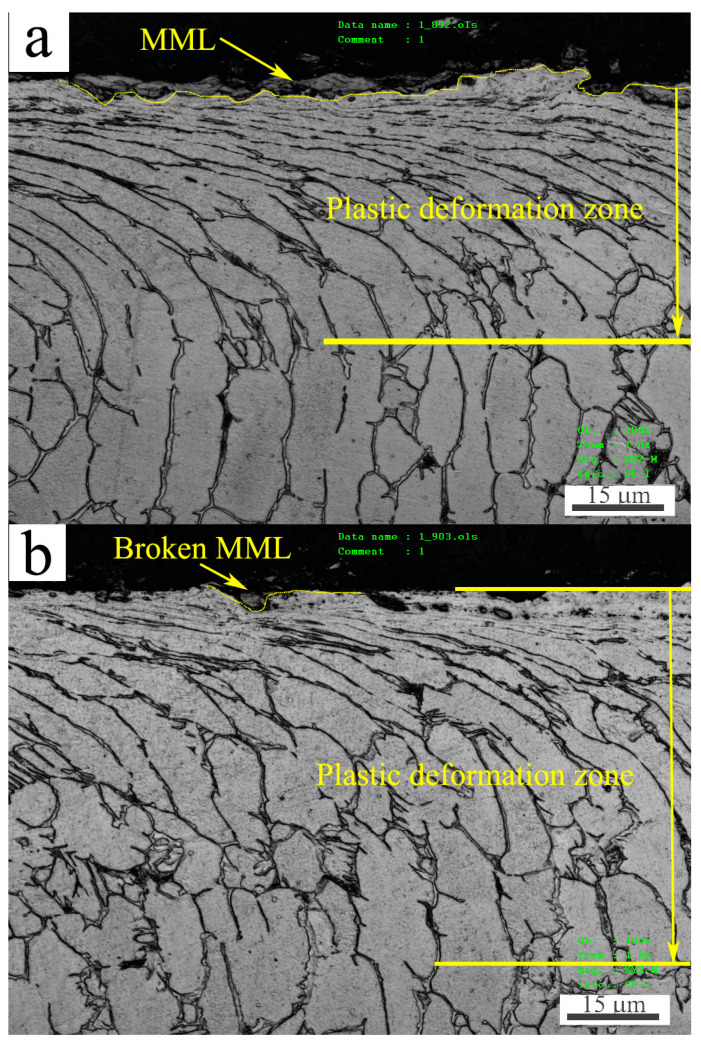
Microstructures in subsurfaces at 50 °C (**a**,**b**) and 150 °C (**c**–**f**) under different loads: (**a**) 100 N, (**b**) 160 N, (**c**) 60 N, (**d**) 60 N, (**e**) 150 N, (**f**) 150 N.

**Figure 10 materials-15-01416-f010:**
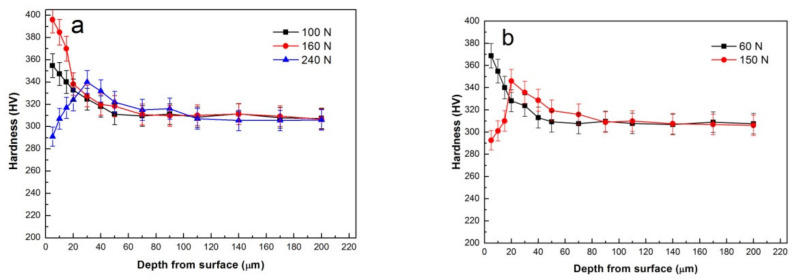
Variation of hardness with depth away from surface under different sliding conditions: (**a**) 100, 160 and 240 N at 50 °C, (**b**) 60 and 150 N at 150 °C.

**Figure 11 materials-15-01416-f011:**
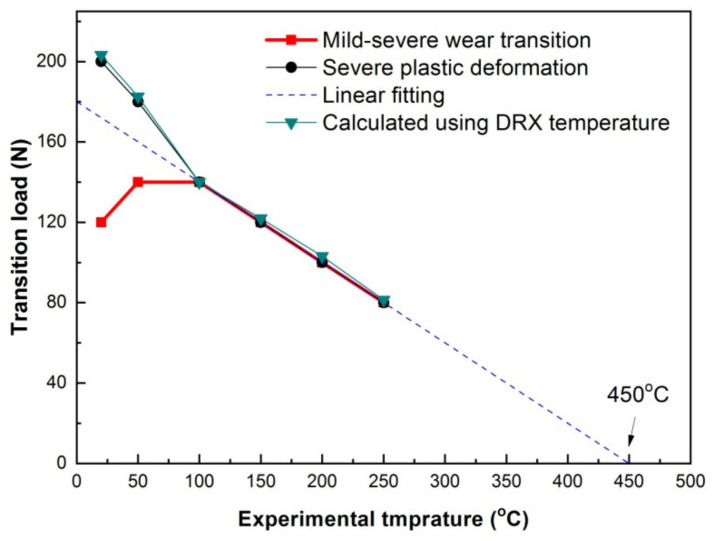
Measured and calculated loads for severe wear transition and SPD wear transition.

**Table 1 materials-15-01416-t001:** Contents of major elements on the worn surfaces at 20 and 50 °C (wt.%).

Temperature (°C)	Load (N)	O	Al	Ti	Fe
20	20	8.89	5.91	79.36	4.16
	80	9.08	5.91	79.29	4.22
	160	8.31	5.84	79.83	4.39
	240	8.57	6.01	80.07	3.82
50	20	8.61	5.80	79.25	4.87
	60	9.31	5.88	77.71	5.74
	100	12.62	5.01	70.63	10.34
	120	10.49	4.90	74.28	8.97
	140	9.17	5.75	78.33	5.10
	160	8.10	5.41	78.36	6.64
	180	7.36	6.07	82.78	2.22
	240	3.38	6.22	87.66	1.11

**Table 2 materials-15-01416-t002:** Wear mechanisms operating at 20 and 50 °C in the first and second stages.

Temperature (°C)	Stage	Load Range (N)	Wear Mechanism
20	First	10–60	Oxidation + abrasion
		80–120	Spallation of MML
	Second	140–200	MPD + Breakdown of MML
		220–240	SPD
50	First	10–60	Oxidation + abrasion
		80–140	Spallation of MML
	Second	160–180	MPD + Breakdown of MML
		200–240	SPD

**Table 3 materials-15-01416-t003:** Contents of major elements on the worn surfaces at 150 and 200 °C (wt.%).

Temperature (°C)	Load (N)	O	Al	Ti	Fe
150	20	12.39	5.41	74.31	6.60
	60	11.01	5.68	75.21	6.71
	100	11.94	5.84	76.96	3.84
	120	11.81	5.41	76.81	4.51
	140	5.84	6.38	85.77	0.63
	150	4.97	5.53	78.99	8.80
200	20	13.43	4.63	67.08	13.74
	60	6.13	5.85	83.18	3.52
	100	9.07	5.29	80.25	3.99
	130	2.04	6.41	88.87	0.96
	140	7.33	5.79	84.20	1.33
	180	7.48	5.90	84.96	0.44

**Table 4 materials-15-01416-t004:** Coefficients of friction at transition loads for severe wear and SPD wear.

Temperature (°C)	20	50	100	150	200	250
Transition load (N)	200 (SPD)	180 (SPD)	140	120	100	80
Coefficient of friction	0.55	0.57	0.65	0.64	0.63	0.64

## Data Availability

Not applicable.

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
