# Peer review of "Two Types of Wear Mechanisms Governing Transition between Mild and Severe Wear in Ti-6Al-4V Alloy during Dry Sliding at Temperatures of 20–250 °C"

_materials, 2022, doi:10.3390/ma15041416_

Round 1

Reviewer 1 Report

The paper concerns chemistry of materials, namely Ti-based alloys and their tribological properties. Due to the wide application of titanium alloys this is a very important topic deciding on potential application and limitations of these materials. Hence, the topic is very interesting and important falling into the scope of the journal. The literature is based on new citations coming from last three years. Hence, the state-of-the-art knowledge in the field is presented. This methods selection is proper allowing for characterization of the materials and their studying. However, this manuscript possess some faults which in my opinion prevent me to accept this paper in the current form. However, taking into account well designed set of experiments and valuable results I propose major revision.

First, English requires significant polishing. Some examples are presented below:

Line 15: “an experimental” instead of “a experimental”

Line 19: ‘tapes”, probably should be “types”

Line 43: “The poor show of titanium alloys…” I d not understand this sentence. Please, rephrase it.

Line 51: “spallationg” better “spallation”

Line 153: “approximately” would be better than “approximate”

Line 351: “express” should be “expressed”

Experimental part is well presented (e.g. wear tests description) but I could not find any details about XRD experiments. Please, provide details of your diffraction experiments.

Line 144: You wrote that “The parts of wear-rate curves within 10-120 N at 20oC…were defind as the first stages” Why? It is not so distinctive this plot for 20 deg as it is for 50 deg. (for 50 deg I completely agree)

Fig. 3g: label on the picture – “deformation”

The phase analysis presented on Fig. 4 is quite convincing apart from magnetite which shows peaks of very low intensity (at the level of background). I give below peaks for 00-001-1111 reference pattern.

No.    h    k    l      d [A]     2Theta[deg] I [%]  

  1    1    1    1      4.85000    18.277       6.0

  2    2    2    0      2.97000    30.064      28.0

  3    3    1    1      2.53000    35.452     100.0

  4    2    2    2      2.42000    37.121      11.0

  5    4    0    0      2.10000    43.038      32.0

  6    4    2    2      1.71000    53.547      16.0

  7    5    1    1      1.61000    57.168      64.0

  8    4    4    0      1.48000    62.728      80.0

  9    6    2    0      1.33000    70.785       6.0

 10    5    3    3      1.28000    73.997      20.0

 11    4    4    4      1.21000    79.079       5.0

 12    6    4    2      1.12000    86.907      10.0

 13    7    3    1      1.09000    89.934      32.0

 14    8    0    0      1.05000    94.381      10.0

At 35 deg the peak can be explained by titanium. Two peaks at 43 and 62 deg are extremely weak and above 70 deg the peak does not correspond neither to 74 nor 79 deg (it is at ca. 77 deg). Can you provide code/number of your reference pattern and is it possible to repeat this experiment because the spectrum is important as you extensively comment these results but this diffractogram shows high noise. Can you extend exposure time to improve signal to noise ratio?

Line 222: “100-140 N” between 100 and 120 the increase seems to be rather mild to me. Can you explain your statement?

Table 3: “0.63” it is rather exceptional result and significant drop in the Fe content followed by significant increase at 150 N. Can you comment those results?

Line 259: “which is decided” what do you mean?

Line 354: “Therefore, Ti-6Al-4V substrate is strain hardened underneath MML” It is known that XRD is good technique for strain studies. Did you consider this method in this project or in the future? See e.g. Yusheng Zhao* and Jianzhong Zhang Journal of Applied Crystallography, (2008). 41, 1095–1108 doi:10.1107/S0021889808031762

References:

Why some of them are given in bold?

Reviewer 2 Report

There are no fundamental remarks on the content of the article. There are comments on the sequence of presentation and design:

  1. In the "Abstract" section, it is not necessary to abbreviate the terms used. All term conventions should be cited as they are used in the "Introduction" and "Experimental Details" sections. Another option would be to introduce a section like "Term Abbreviations". Then it will not be necessary to repeat their decoding in the text of the article and in the captions. Also, do not use abbreviations in the "Conclusions" section, as this will be very inconvenient for readers of the article.
  2. In the “Experimental Details” section, it is necessary to separate subsections with a description of methods for measuring hardness and conducting studies of friction surfaces, microstructure. Also, for the convenience of perceiving the text of the article in the “Wear Tests” subsection, it is necessary to describe in more detail the wear rate indicator used by the authors in the form of volume loss of material per unit of sliding path, indicating the unit of measurement of this indicator.
  3. From the titles of the subsections "Wear Behavior at 20 and 50 oC" and "Wear Behavior at Temperatures of 100-250 oC" it is desirable to remove the word "Behavior", since they discuss not only the wear rate or coefficient of friction, but also the morphology and chemical composition worn surface, as well as wear mechanisms.
  4. In the "Conclusions" section, you can recommend reducing the number of very short conclusions by combining them into more meaningful ones.

Reviewer 3 Report

The authors present results from pin-on-disc studies of Ti-6Al-4V at various applied loads and operating temperatures.  Wear measurements are supplemented with hardness data and SEM/EDS observations.  The data presentation is generally clear, but the conclusions are not always consistent with the data as presented.  Some distinctions (e.g. the onset or definition of 'sever wear') appear to be arbitrarily made.  While the paper is understandable, a complete review for English readability is also advised.

Lines 100-104: please cite the method used to measure the grain size (standard used or technique employed); what was average width of grains?

Line 132: what does 'vertical to the worn surface' mean?  I think it is implying a vertical distance from the wear surface but am not sure.

Line 142 - it is difficult to tell from the graph, but it appears that the 50C/120N data point is higher than the value at 80N (outside of experimental uncertainty; the 140N point definitely is - the statement about 'except for a plateau with lower wear rate' does not seem to be justified; the range from 160N-220N has a longer force range with statistically similar values of wear rate; the 80N-140N segment is not the only unique feature

Lines 217-219: there does not seem to be an overall change in slope in Figure 5 corresponding to 'wear rates...accelerate with increasing load'; there are possibly additional plateaus, the but relationship between wear rate and load appears linear for all temperatures over the range of data presented

Figure 7: does 'severe wear' correspond to a specific wear rate or is it intended to represent a specific mechanism of wear?  For example, from Figures 1 and 5, it appears that a load of 140N (shown as the boundary with severe wear in Figure 7) corresponds to wear rates of 30 and 35 for temps of 50C and 100C, respectively - a 15% difference.  If there is a reason that wear mechanism is more important than wear rate for users in the definition of 'severe', the authors need to explain more thoroughly.  The definition of a wear map with a severe wear transition is somewhat misleading because the wear rates increased basically linearly with load.

Figure 10: the results for 240C and 150C do not appear to be consistent with the data in Figure 8; no hardness measurements in Figure 8 are lower than about 380HV; Figure 10 has measurements at the surface of 290-300HV - what is the discrepancy?  The lower values are explained in the text as corresponding to recrystallization, but that doesn't explain why the lower values weren't measured at the surface for higher temperatures; there is not a clear surface structure in Figure 9f that would justify a significantly higher measurement right at the surface

Figure 11: it is unclear from the presented data why the dynamic recrystallization temperature should somehow play a role in the wear rates at temperatures when dynamic recrystallization is not observed and the wear mechanisms are significantly different; more explanation of this hypothesis is required; only one image (and two sets of hardness measurements) was presented that showed any evidence of dynamic recrustallization; in addition, no dynamic recrystallization was claimed to be observed at 100C

Lines 429-432: there is little/no support in the data provided for the statement that a 'rapidly rising stage' exists; there is a single data point at the end of most wear graphs that may show a jump but, based on the rest of the data set, it is equally likely that this could be the start of another plateau - regardless, the authors claim the transition to the 'rapidly increasing stage' actually occurred at a much lower load and this is not justified by the data

Round 2

Reviewer 1 Report

I am glad that this manuscript is significantly improved and its significance increased.  However, I vote for minor revision because XRD experiments are not described properly. I would like to ask you for information about angle range, angle step and exposure time applied in the experiment. I would like also to ask you for presenting (only to me and not in the manuscript) of magnetite diffractogram separately and not imposed with titanium phases as you wrote:

"The peaks of magnetite are at low intensity as compared with the peaks of a-Ti, b-Ti and TiO phase, but they are can be identified in their respective XRD patterns at 20oC and 50oC. When two sets of XRD patterns are put in one figure, the peaks of magnetite are not well recognized."

Reviewer 3 Report

I am satisfied with the author's response and changes.  A few minor English changes:

Line 85 - I think 'to' should be 'too'

Line 271 - 'form' should be 'from'

Line 304 - 'is comprised by' should be 'comprises'

Line 362 - 'soften' should be 'softening'

Line 497 - 'increase' should be 'increasing'
